# A Comprehensive Review of Advanced Lactate Biosensor Materials, Methods, and Applications in Modern Healthcare

**DOI:** 10.3390/s25041045

**Published:** 2025-02-10

**Authors:** Yifeng Ding, Liuhong Yang, Jing Wen, Yuhang Ma, Ge Dai, Fengfeng Mo, Jiafeng Wang

**Affiliations:** 1School of Health Science and Engineering, University of Shanghai for Science and Technology, 516 Jungong Road, Shanghai 200093, China; 2Department of Anesthesiology, Changhai Hospital, Naval Medical University, Shanghai 200433, China; 3Department of Naval Nutrition and Food Hygiene, Faculty of Naval Medicine, Naval Medical University, 800 Xiangyin Road, Shanghai 200433, China; 4Key Laboratory of Biosafety Defense, Naval Medical University, Ministry of Education, 800 Xiangyin Road, Shanghai 200433, China

**Keywords:** lactate, biosensor, electrochemical, medical, enzyme

## Abstract

Lactate is a key metabolite in cellular respiration, and elevated levels usually indicate tissue hypoxia or metabolic dysregulation. The real-time detection of lactate levels is particularly important in situations such as exercise, shock, severe trauma, and tissue injury. Conventional lactate assays are insufficient to address today’s complex and variable testing environments, and thus, there is an urgent need for highly sensitive biosensors. This review article provides an overview of the concept and composition of electrochemical lactate biosensors, as well as their recent advances. Comparisons of popular studies on enzymatic and non-enzymatic lactate sensors, the surface-related materials used for modifications to electrochemical lactate biosensors, and the detection methods commonly used for sensors are discussed separately. In addition, advances in implantable and non-implantable miniaturized lactate sensors are discussed, emphasizing their application for continuous real-time monitoring. Despite their potential, challenges such as non-specific binding, biomaterial interference, and biorecognition element stability issues remain during practical applications. Future research should aim to improve sensor design, biocompatibility, and integration with advanced signal processing techniques. With continued innovation, lactate sensors are expected to revolutionize personalized medicine, helping clinicians to increase treatment efficiency and improve the experience of their use.

## 1. Introduction

Lactate, as a key intermediate product in cell metabolic processes, has long been regarded as an important indicator for evaluating human physiological states and disease diagnoses. In anaerobic conditions, lactic acid is produced by cells through the glycolytic process to maintain a certain level of energy supply [1]. The normal level of lactate in the human bloodstream is usually between 0.5 and 1.5 mmol/L, but in situations such as exercise, shock, severe trauma, and tissue damage, lactate levels can rise to 5–15 mmol/L [2,3,4,5,6]. In conditions such as mild physical activity or stress, lactate levels can rise to 2.0–4.0 mmol/L, which is still considered to be manageable [7]. However, during intense exercise, muscle lactate may locally accumulate to levels as high as 15–20 mmol/L, far exceeding normal blood concentrations [8]. In the medical field, the abnormal elevation of lactate levels may indicate tissue hypoxia or metabolic dysregulation, which are considered to be unfavorable signs in various clinical situations [9]. Therefore, the real-time and accurate monitoring of lactate levels is crucial for clinicians, as it not only helps doctors to assess the severity of a condition, but also can be used to monitor changes in such conditions and the effectiveness of treatment [10].

Lactate monitoring plays a vital role across various medical fields, such as critical care management, perioperative monitoring, and sports medicine [11,12,13]. For critically ill patients in the intensive care unit (ICU), who are often under high stress, their energy metabolism accelerates, causing an increase in lactate levels [14]. Lactate levels of ≥2 mmol/L are a key diagnostic criterion for septic shock, correlating with severity and prognosis [12]. The continuous monitoring of lactate levels is crucial for the early detection of life-threatening conditions such as infections, septic shock, and tissue hypoxia, where lactate levels of ≥4 mmol/L indicate lactic acidosis [15]. The early identification of these conditions allows healthcare providers to intervene promptly, improving patient outcomes. Especially during major and cardiac surgeries, real-time lactate monitoring helps to evaluate patients’ metabolic status and tissue perfusion [16,17]. Significant lactate changes during surgery may signal blood flow deficiency or tissue hypoxia, and the early detection of these signs is crucial for preventing postoperative complications [18]. Rapid lactate measurements are essential for diagnosing and treating patients suspected of shock, severe infection, or acute metabolic disorders [19,20,21].

For trauma patients, lactate levels of ≥4 mmol/L indicate inadequate tissue perfusion, a condition closely associated with higher mortality rates [22]. In emergency patients, lactate levels of ≥8 mmol/L are associated with a high 30-day mortality risk [23]. Real-time data from lactate sensors accelerate diagnostic processes, allowing for quicker medical interventions. At the same time, lactate monitoring aids in assessing the physical condition and training intensity of athletes in sports medicine [24]. Real-time lactate level monitoring facilitates the optimization of training regimens, effectively mitigating risks of overtraining and musculoskeletal injuries. Similarly, lactate monitoring proves valuable in managing chronic diseases, where fluctuating lactate levels can indicate disease progression and treatment effectiveness, especially in conditions like diabetes, cardiovascular diseases, and cancer [25,26,27,28]. Due to the high metabolic activity of cancer cells, the lactate levels in the tumor microenvironment can reach 10–30 mmol/L, significantly higher than normal levels [29]. The real-time monitoring of lactate levels can provide clues about tumor status and treatment response. Providing a comprehensive overview of lactate levels under various physiological and pathological conditions, Table 1 summarizes blood lactate concentration ranges, encompassing states from normal metabolism to critical illness. The table highlights the diagnostic and prognostic significance of lactate, offering valuable insights into its dynamic fluctuations and their close association with the metabolic and health status of the human body in both clinical and research contexts.

In this paper, we briefly describe the recent developments and applications of lactate electrochemical biosensors. In addition to this, the article summarizes the differences between lactate electrochemical biosensors in terms of materials, detection methods, applications, and future directions, with a special focus on how the fundamentals of detection methods and material innovations can enhance sensor performance. In addition, we critically evaluate existing lactate sensor reviews (Table 2) and analyze their strengths and limitations, thus clarifying the direction of our review. On this basis, we choose to delve into the comprehensive applications of lactate electrochemical sensors in clinical diagnosis, intensive care, and chronic disease management from the perspective of their fundamental detection principles and materials. At the same time, we summarize the advantages and disadvantages of current lactate electrochemical sensors and propose directions for future research to address existing challenges. The following figure (Figure 1) demonstrates the working principle of lactate electrochemical sensors.

## 2. Techniques for Lactate Detection in the Medical Domain

Currently, in the biomedical field, the main methods for lactate determination are high-performance liquid chromatography (HPLC), fluorescence, colorimetric, chemiluminescence, and magnetic resonance spectroscopy. In the laboratory, Enzyme-Linked Immunosorbent Assay (ELISA) kits are used as a portable means for detecting lactic acid. Lactic acid is converted into pyruvate by lactate dehydrogenase (LDH), with the simultaneous production of reduced coenzyme I (Nicotinamide adenine dinucleotide, NADH). The absorbance of NADH can be measured at 340 nm to calculate the lactate content of blood [37]. In addition to ELISA kits, alternative methods such as lactate test strips and HPLC can be used to detect lactate. However, lactate test strips are only suitable for initial screening and are not applicable for the precise measurement of lactate concentrations in the medical field, so their accuracy and reproducibility need to be further improved [38]. HPLC is a widely used analytical technique in laboratories that can be utilized for the quantitative determination of lactate in complex samples [39]. When complex mixtures (plasma, serum, and tear fluid) are involved, HPLC can effectively separate lactic acid from other interfering substances and is, therefore, particularly suitable for the accurate quantification of lactic acid [40]. However, its application in the medical field is limited by the need for expensive instrumentation costs, complex operational procedures, and time-consuming pre-treatment. Instead, enzyme-based spectrophotometric methods are mainly used to detect lactate concentrations in clinical practice. Lactic acid is catalyzed by LDH to produce NADH, which is subsequently detected spectrophotometrically in a specific wavelength band, yielding the corresponding lactic acid concentration in the sample. The current is captured by the sensor circuitry and converted into a lactate concentration. However, this method, when applied in the medical field, not only needs a shorter processing time, but also does not allow for the continuous detection and real-time monitoring of lactate in patients in the clinic.

In contrast, the emergence of electrochemical biosensor methods has brought about a new solution. Following the discovery of the first enzyme-based electrochemical biosensor for glucose by Clark and Lyons in 1962, electrochemical biosensors have been widely used in the field of medical diagnostics detection due to their advantages of rapid detection, ease of use, low cost, and less pre-processing [41]. The working principle of electrochemical biosensors is to capture electrical signals (for example, voltage, current, and impedance) by studying the chemical reactions occurring on the surface of the electrodes of a substance, to realize the detection and quantification of the substance to be tested. Electrochemical sensors for the detection of lactic acid usually use enzymes, nanomaterials, nano-enzymes, or molecularly imprinted polymers (MIPs) that react specifically to lactic acid as a biometric element, thereby greatly improving the sensitivity and specificity of such sensors [42]. Electrochemical biosensors have a wide range of applications in the medical field due to their high sensitivity, high accuracy, affordable price, sustainable measurement, miniaturization, and portability.

## 3. Biometric Elements for Lactic Acid Electrochemical Sensors

As an important component of electrochemical biosensors, biorecognition elements can greatly affect the specificity and sensitivity of electrochemical sensors. Because biorecognition elements each have different physicochemical properties, they also affect the choice of materials for the subsequent transduction layer when composing electrochemical biosensors. In recent studies on lactate electrochemical biosensors, enzymes [43], metal oxides, nanoparticles [44], metal–organic frameworks [45], and MIPs [46] have all been reported as biorecognition elements for sensors.

### 3.1. Enzyme-Based Lactate Electrochemical Biosensors

Enzyme-based electrochemical sensors combine enzyme specificity with electrochemical transduction principles to detect and quantify specific analytes (Figure 2). The selection of enzymes during sensor construction ensures the feasibility and specificity of the sensor.

Electrochemical sensors for the detection of lactic acid usually use enzymes that react specifically to lactic acid as the core biorecognition element to achieve the specific detection of lactic acid. Lactate oxidase (LOx) is one of the core recognition molecules for the detection of lactate in enzyme electrode electrochemical sensors. Its use in the detection of lactate involves catalyzing the production of pyruvate and hydrogen peroxide promoted by lactate. Hydrogen peroxide is oxidized and decomposed on the electrode surface to produce a current proportional to the lactate concentration (Equation (1)) [47]. A wearable lactate biosensor based on lactate oxidase(Sigma-Aldrich Co. LLC, St. Louis, MO, USA) was reported by Jiang et al. [48]. The sensor was based on the layer-by-layer in situ deposition of Prussian blue (PB) and reduced graphene oxide (rGO) with sea urchin-like gold nanoparticles as a bracket to fix LOx. It also achieved a high sensitivity of 40.6 µA mM^−1^ cm^−2^ in the range of 1–222 µM and 1.9 µA mM^−1^ cm^−2^ in the wider range of 222 µM–25 mM. It provides the stable long-term monitoring of lactic acid in sweat and is suitable for real-time applications in fields such as sports and healthcare. LDH is also used to catalyze lactate at the electrode surface to promote the production of pyruvate and NADH from lactate (Equation (2)) [49,50]. The electrochemical oxidation of NADH to its corresponding oxidized form (NAD+) in aqueous solution under the applied voltage of a sensor generates a current proportional to the lactate concentration [50,51]. A paper by Chan et al. described a dual-enzyme biosensor for lactate detection. Lactate dehydrogenase (LDH)(Sigma-Aldrich Chemie GmbH—Schnelldorf, Germany) and pyruvate oxidase (PyrOx) (Sigma-Aldrich Chemie GmbH—Schnelldorf, Germany) were immobilized separately on screen-printed carbon electrodes (SPCEs), which were simultaneously cross-linked using glutaraldehyde vapor. The sensor measures lactate by monitoring the impedance changes caused by ions produced by the enzyme-catalyzed reaction. It has a detection limit of 17 µM by an impedance-based test method, can achieve a linear range of 10 µM–250 µM, and has a high operational stability for up to one month (Figure 3A) [49].(1)$$\begin{array}{c} \mathrm{Lactate}+O_{2}\overset{\mathrm{LO}x}{\to }\mathrm{Pyruvate}+H_{2}O_{2}\\ H_{2}O_{2}\to O_{2}+2H^{+}+2e^{-} \end{array}$$(2)$$\begin{array}{c} \mathrm{Lactate}+NAD^{+}\overset{LDH}{\to }\mathrm{Pyruvate}+NADH+H^{+}\\ NADH\to NAD^{+}+H^{+}+2e^{-} \end{array}$$

### 3.2. Enzyme-Free Lactate Electrochemical Biosensors

Although sensors containing enzymes have a good specificity, on the other hand, enzyme activity can be affected by factors such as temperature, pH, and chemicals. This leads to fluctuations in stability and reproducibility during the actual detection process. Therefore, researchers have sought to use other materials with electrocatalytic activity as the core recognition element in electrochemical lactate sensors. These materials offer unique advantages in lactate sensor design, making them suitable alternatives to enzymes. Enzyme-free lactate sensors utilize materials with electrocatalytic activity to replace traditional enzymes as recognition elements. These electrocatalytic materials provide unique advantages in lactate sensor design and serve as effective substitutes for enzymes. They allow for an enhanced sensor stability, increased reproducibility, and better performance under varying environmental conditions.

Metal oxides like CuO [52], ZnO [53], NiO [54], and TiO_2_ [55] are widely used in enzyme-free electrochemical lactate sensors due to their excellent catalytic properties and large surface areas. These materials facilitate efficient lactate oxidation by promoting electron transfer and enhancing stability, even in extreme conditions. The enzyme-free electrochemical lactate sensor reported by Wu et al. uses bimetallic nickel-based layered double hydroxide (Ni LDH) for the enzyme-free electrochemical detection of lactate (Figure 3B) [44]. The sensor’s high selectivity was achieved by leveraging the unique electrochemical properties of Ni LDH, which preferentially facilitates the oxidation of lactate over other potential interfering substances. This selectivity was demonstrated both in blank solutions and in the presence of common interferents such as glucose and ascorbic acid, ensuring accurate lactate detection in complex sample matrices without significant interference from other metabolites. Co metal was used as a second metal to enhance the electrocatalytic activity and lactate oxidation. Nanoparticles, especially plasmonic gold nanoparticles (AuNPs) [56,57] and platinum nanoparticles (PtNPs) [58], are often used to improve sensor performance. Their superior electrical conductivity and larger surface area allow for stronger catalysis with lactates, resulting in an increased sensitivity, lower detection limits, and faster response times. Metal–organic frameworks (MOFs) are also widely used in enzyme-free sensing due to their highly porous structure and tunable properties. Due to their large surface area and customizable pore size, MOFs such as Cu-MOFs [59] and ZIF-67 MOFs [45] are useful for the detection of lactate under various physiological conditions. Wang et al. developed a non-enzymatic electrochemical lactate sensor using bimetallic NiCo layered double hydroxide (NiCo-LDH) derived from ZIF-67, optimized for the analysis of human sweat (Figure 3C) [45]. The layered structure of NiCo-LDH enhances the surface response area of the sensor, which allows for the stable and reproducible detection of lactic acid in sweat samples under different physical activity conditions. Finally, MIPs offer a high specificity by mimicking natural recognition elements, creating selective cavities for lactate binding. Zhang et al. presented the development of a wearable, non-invasive lactate sensor using MIPs combined with silver nanowires (AgNWs) for the electrochemical detection of lactate in human sweat (Figure 3D) [46]. MIPs were electropolymerized with 3-aminophenylboronic acid on AgNW-coated electrodes to create recognition sites specific for lactate. MIPs improve sensor durability and allow for selective detection in complex samples, making them suitable for long-term clinical and environmental monitoring.

The integration of these materials into enzyme-free electrochemical lactate sensors is more conducive to the development of more durable and stable devices. These materials, which can catalyze lactate independently of enzymes, help to improve cost-effectiveness, ensure a better and more stable real-time performance of sensors, and make it easier to implement sensors for subsequent continuous monitoring applications in medical diagnostics, sports performance, and industrial processes.

## 4. Electrochemical Lactate Sensors with Different Modified Materials

The choice of modified materials also has a critical impact on the performance of lactate electrochemical biosensors. The performance enhancement of lactate electrochemical biosensors can also be realized by the modification of common materials. Common material modifications (such as nanomaterials, ion-selective osmotic membranes, polymers, and hydrogels) have often been applied to sensors in recent years and have played an important role in improving the performance and functionality of lactate biosensors.

### 4.1. Lactate Electrochemical Biosensors Based on Nanomaterial Modifications

Nanomaterial modifications are widely used in lactate electrochemical biosensors due to their excellent electrocatalytic performance and high surface area. Common nanomaterials include metal nanoparticles [60], carbon nanotubes [61,62], graphenes [63], MOFs [64], and MXenes [65]. Pereira et al. used AuNPs to modify a glassy carbon electrode along with molecularly imprinted polymers to construct a lactate electrochemical sensor. This modification significantly enhanced sensitivity and facilitated more efficient electron transfer [66] (This summary chart is detailed in Figure 4A at the end of this section). The molecularly imprinted polymers (MIPs) specifically tailored for lactate molecules enhanced the sensor’s selectivity by providing recognition sites that only bound lactate, even in the presence of common interfering substances such as glucose and uric acid. The high specificity of the MIPs ensured an excellent selectivity under both blank conditions and in the presence of potential interferents, making the sensor highly reliable for lactate detection in complex biological samples. Tian et al. developed a lactate sensor by immobilizing lactate oxidase and horseradish peroxidase on NiCo_2_O_4_ microspheres and single-walled carbon nanotubes (SWCNTs). It achieved a wide linear detection range of 0.1–30 mM [62]. Jiang et al. developed a lactate biosensor by using a zinc oxide–graphene oxide nanocomposite for the detection of lactate levels in saliva samples, with a detection limit reaching 9 µM [53].

### 4.2. Membrane-Based Lactate Electrochemical Biosensors

A membrane is a selective barrier that regulates the passage of ions, molecules, or particles in and out of a system. In electrochemical biosensors, membranes play a crucial role by providing selective permeability, ensuring that only the target analyte, such as lactate, passes through, while blocking interfering substances [67]. Common membrane materials include nanoporous membranes, ion exchange membranes, and molecular sieve membranes. Nanoporous membranes selectively allow lactate molecules to pass while blocking interfering substances [68]. Wu et al. developed a bifunctional wearable lactate biosensor using a nanoporous polycarbonate (PC) membrane, which exhibited a sensitivity of 82.4 nA mM^−1^ with minimal high sensitivity, capable of real-time dual-parameter detection of sweat lactate and temperature [69] (This summary chart is detailed in Figure 4B at the end of this section). Scanning Electron Microscopy (SEM) was used to characterize the morphology and elemental composition of gold-modified PC nanoporous membranes, confirming the homogeneous distribution of nanostructured Au clusters, which led to an improved electron transfer and detection performance. The enzyme has been encapsulated in an etched PC nanoporous electrode, while CS hydrogel provides an ideal encapsulation matrix for the enzyme on this PC nanoporous electrode. This design ensures the enzyme’s specificity and long-term stability. Reza et al. used a Nafion membrane to cover a microneedle sensor for the detection of lactate. The Nafion membrane improved the selectivity and stability of the sensor, which can achieve a sensitivity of 43.96 µA mM^−1^ cm^−2^ [70]. A PB/rGO composite membrane, fabricated through an in-situ layer-by-layer spin-coating process, can ensure specificity by targeting the reaction pathway of immobilized lactate oxidase (LOx). This targeted immobilization enables the membrane to catalyze lactate with a high selectivity [48]. In the presence of interfering substances such as uric acid, ascorbic acid, and acetaminophen, the sensor can achieve a 2–4% accuracy compared to individual tests.

### 4.3. Polymer-Based Lactate Electrochemical Biosensors

Due to their good conductivity and tunable chemical properties, polymers are widely used in lactate electrochemical biosensors. Common polymers include polypyrrole (PPy), polyaniline (PANI), and polyvinyl alcohol (PVA). Zhang et al. developed a lactate electrochemical sensor using a composite of PPy and multi-walled carbon nanotubes (MWCNTs). The PPy/MWCNT composite enhanced the electrical properties and electrochemical activity of the sensor, significantly improving its sensitivity and stability during lactate detection [71]. Mugo et al. developed a lactate electrochemical sensor using a PANI and 3-aminophenylboronic acid co-polymerized membrane embedded with AuNPs on an SPCE (This summary chart is detailed in Figure 4C at the end of this section). The PANI-based membrane significantly enhanced sensitivity and selectivity during lactate detection by providing a robust and conductive interface [72].

### 4.4. Hydrogel-Based Lactate Electrochemical Biosensors

Hydrogels are widely used in lactate electrochemical biosensors due to their high water content and excellent biocompatibility. Common hydrogel materials include PVA hydrogels, polyacrylamide (PAM) hydrogels, and sodium alginate (SA) hydrogels [73]. Phumma et al. embedded a silver nanoparticle and nanocellulose (HNC/AgNP) composite into a PVA hydrogel to fabricate a lactate electrochemical sensor. The PVA hydrogel matrix provided a stable and conductive environment, significantly enhancing the sensitivity and stability for lactate detection [74]. Mukundan et al. developed a lactate electrochemical sensor using a bimetallic Fe/Co-MOF embedded in a PVA/chitosan hydrogel matrix, which significantly enhanced the sensitivity and durability for sweat-based lactate detection, with a detection limit of 10 nM [75]. Somchob et al. developed an electrochemical lactate biosensor by immobilizing lactate oxidase on a nitrogen-doped graphene electrode, using a co-polymer poly [2-methacryloyloxyethyl phosphorylcholine (MPC)-co-N-methacryloyloxyethyl tyrosine methylester (MAT)] (PMM) hydrogel to enhance the enzyme stability and activity over an extended period (This summary chart is detailed in Figure 4D at the end of this section). The sensor retained 80% of its activity over 49 days, while achieving a wide detection range of 0–25 mM (R^2^ = 0.9661) [76].

**Figure 4 sensors-25-01045-f004:**
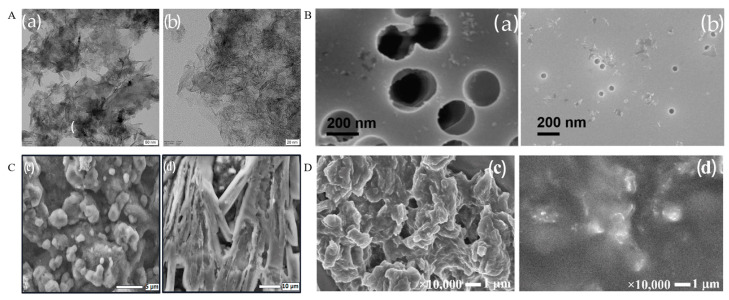
(**A**) SEM images of RGO/GCE (**a**) and AuNPs/RGO/GCE (**b**) [66]; (**B**) SEM images of the etched PC nanoporous membrane electrode [69]; (**C**) SEM images of the epinephrine MIPs SPCE sensor and lactate MIPs SPCE sensor [72]; and (**D**) SEM images of SPCE/NDG/Gox and SPCE/NDG/GOx/PMM70 [76].

In electrochemical lactate sensors, different types of materials exhibit distinct advantages and disadvantages. Based on material categories, we can summarize some common features. Nanomaterials (such as AuNPs, CNTs, graphene, MOFs, and MXene) typically possess a high surface area, excellent conductivity, and enhanced catalytic activity, characteristics that significantly improve the sensitivity and response time of electrochemical sensors. However, the drawbacks of nanomaterials primarily relate to their dispersion performance. For instance, AuNPs tend to aggregate during the modification process, and CNTs may disperse unevenly on the electrode, potentially affecting their performance. Additionally, these materials suffer from a high cost and complex fabrication processes, which are their main limitations. Membrane materials, such as Nafion and PC, enhance sensor performance by providing selective permeability and protecting enzyme activity. Nafion membranes are widely used due to their high ionic conductivity and selective permeability, but their high cost and relatively poor conductivity in certain applications are significant drawbacks. In contrast, PC membranes offer a good mechanical strength and stability, but they face challenges such as an inability to completely block small molecule interferents. Polymer materials, including PPy, PANI, and PVA, generally offer a good conductivity and tunable chemical properties, which enable them to provide an excellent electrochemical activity and good biocompatibility in sensors. However, polymers tend to have a poor mechanical strength, especially when exposed to long-term use or large mechanical stresses, leading to brittleness or deformation. Moreover, some polymers are prone to swelling or instability, which can result in performance degradation in practical applications. Additionally, hydrogel materials such as PVA/PAM and SA hydrogels are widely used in lactate electrochemical sensors due to their high-water content and excellent biocompatibility. They provide stable and conductive environments, thus enhancing the sensitivity and stability of sensors. However, these hydrogels often suffer from an insufficient mechanical strength and tendency to degrade, especially in dynamic biological environments, which limits their long-term applicability. The common features and limitations of these materials determine their various applications in electrochemical lactate sensors. The advantages and disadvantages of the materials mentioned in the referenced articles are summarized in Table 3.

## 5. Lactate Electrochemical Sensors with Different Detection Methods

Signal transduction and amplification play crucial roles in electrochemical biosensors. Once the biorecognition element identifies the target analyte, various electrochemical techniques are employed to convert the resulting chemical changes into the desired electrical signals. These techniques include electrochemical impedance spectroscopy (EIS) [77], differential pulse voltammetry (DPV) [46,68], cyclic voltammetry (CV) [72,78], and amperometry (AMP) [79,80]. These technologies use different electrochemical reaction mechanisms to convert and amplify signals, ensuring detection sensitivity and specificity.

### 5.1. Electrochemical Impedance Spectroscopy (EIS)

By using the EIS technique along with the addition of sensing materials to the electrode surface, researchers have discovered another way to detect lactate. Sensing materials can measure specific changes at the surface interface caused by lactate [58]. The EIS technique can be used to test the resistance of individual materials or entire sensor systems over a range of frequencies. During the measurement process, a sensor is subjected to an alternating current (AC) voltage, and the subsequent current response is measured to calculate the impedance of the sensor system, which provides information about the electrochemical properties of the sensor surface, such as its complex resistance and capacitance characteristics [81]. Lin et al. developed a flexible graphene oxide (GO) to immobilize LOx to construct a lactate biosensor. The sensor utilized EIS to detect the lactic acid in sweat. GO nanosheets significantly improved its sensitivity, with a dynamic range of 1–100 mM and a detection limit of 1 mM [77] (This summary chart is detailed in Figure 5A at the end of this section). Nien et al. developed a flexible arrayed lactate biosensor using copper-doped zinc oxide (CZO) films modified with iron–platinum nanoparticles (FePt NPs). The FePt NPs significantly reduced the sensor’s charge transfer resistance and enhanced its electrochemical activity, improving its sensitivity for lactate detection using EIS [58]. Morrow et al. developed a lactate electrochemical sensor by immobilizing lactate dehydrogenase on a gold disk electrode using an SAM and utilized EIS to measure the lactate concentration, significantly improving detection sensitivity and reducing detection time [30]. EIS also aids in optimizing sensor design, making it a preferred choice for advanced lactate biosensors [82].

### 5.2. Differential Pulse Voltammetry (DPV)

As a highly sensitive electrochemical technique, DPV is widely used to determine the concentration of single or multiple target substances. When a sensor is tested with the DPV technique, a series of short voltage pulses are applied to the electrode, and the resulting current is measured. This method significantly improves the signal-to-noise ratio of the sensor and is, therefore, more commonly used for the detection of low concentrations of lactic acid. Khan et al. fabricated a lactate electrochemical sensor using a modified graphite–polyurethane–rGO–PB (G-PU-RGO-PB) composite on a hydrophobic textile substrate, employing DPV for lactate detection (This summary chart is detailed in Figure 5B at the end of this section). This approach significantly enhanced sensitivity and stability in real-time sweat analysis [83]. Another study showed that a DPV sensor could accurately measure the lactate levels in human sweat, highlighting its potential for non-invasive monitoring [46]. The high sensitivity and specificity of DPV have led to its use as the first choice for clinical diagnostics, environmental monitoring, and industrial applications [84]. By applying the DPV test method, sensors can realize the fast and precise detection of target substances, making it one of the important methods for manufacturing high-sensitivity sensors.

### 5.3. Cyclic Voltammetry (CV)

As an important and widely used technology in the field of electrochemistry, CV is often used to describe the redox behavior of electroactive substances on the surfaces of electrodes. During the test, potential is applied to a sensor and gradually changes to a set endpoint value during a linear scan [78]. The applied point potential is then returned along the reverse direction, forming a cyclic pattern. This test method allows for the oxidation and reduction potentials of the sensor to be determined for later use in understanding reaction mechanisms and optimizing sensor performance [85]. Ben Moussa et al. developed a highly selective electrochemical lactate biosensor by combining modified gold, rGO, and AgNPs with MIPs (AuE/rGO-AgNPs/MIPs) (This summary chart is detailed in Figure 5C at the end of this section). The sensor used CV to detect lactate with a remarkable sensitivity and selectivity. The detection limit of this sensor was 726 nM, with a linear range from 10 to 250 µM [86]. In addition, Sainz et al. developed a lactate biosensor by covalently immobilizing LOx onto chevron-like graphene nanoribbons. The anodic peak current acquired by this sensor when operating at 0.3 V enabled an accurate assessment of the lactate detection response. The biosensor had a detection limit of 11 µM and a linear range of 34–280 µM. It also had a detection limit of 11 µM [87]. The versatility of CV makes it an essential technique for materials research and electrochemical sensor design, allowing researchers to readily adapt it to a wide range of applications during development.

### 5.4. Amperometry (AMP)

Among electrochemical sensors for lactic acid detection, AMP is widely used because of its ease of testing and high sensitivity. During sensor testing, the AMP technique applies a constant potential to the electrode surface and measures the strength of the current transmitted to the electrode surface. The recorded current can be correlated with the volumetric concentration of the electroactive substance or its change in the study sample. Phamonpon et al. developed an electrochemical sensor for lactic acid using PtNP/rGO composites on carbonized silkworm cocoons and detected the lactic acid in sweat using the amperometric method. This method improved the sensitivity of the sensor to 0.87 µA mM^−1^ cm^−2^ with a detection limit of 70 µM [60] (This summary chart is detailed in Figure 5D at the end of this section). Xu et al. developed an in-ear integrated lactate electrochemical sensor using a PB-modified electrode covered with a PVA hydrogel, utilizing chronoamperometry for lactate detection in sweat [88]. This design significantly enhanced sensitivity, maintaining a consistent performance at a sensitivity of −0.47 µA mM^−1^ within a temperature range from 25 °C to 40 °C and a humidity range from 40% to 70%. It also demonstrated stable operation over a continuous 1 h of use. Additionally, amperometric sensors have been employed in the food industry to ensure quality control by detecting lactate in dairy products [89,90,91]. AMP’s ability to provide rapid and accurate measurements enhances its utility in different monitoring scenarios [92,93].

**Figure 5 sensors-25-01045-f005:**
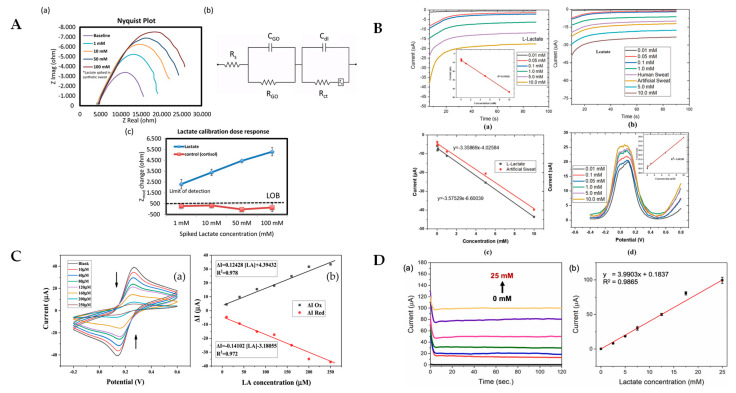
(**A**) Nyquist response plots (**a**), fitted Randles circuits (**b**), and lactate-calibrated dose responses (**c**) for GO/PANHS/LOD lactate sensor at different lactate concentrations were reported by Lin et al. [77]; (**B**) amperometric detection response plots (**a**) and their linear correlation curves (inset) for G-PU-RGO-PB/LOx lactate sensor at different lactate concentrations, amperometric detection response plots (**b**) and their linear correlation curves (**c**) for human and artificial sweat, and differential pulse voltammograms (**d**) including linear calibration curves (inset) at different lactate concentrations, as reported by Khan et al. [83]; (**C**) cyclic voltammetry response plots obtained by the CV method of AuE/rGO-AgNPs/MIPs lactate sensor for detection between −0.2 and 0.6 V at lactate concentrations ranging from 0 to 250 µM lactate, and their linear correlation plots of the current variations of the anodic (**a**) and cathodic (**b**) peaks, as reported by Ben Moussa et al. [86]; and (**D**) amperogram (**a**) and calibration plot (**b**) between versus current response obtained for 0 to 25 mM lactate concentration by LOx/PtNPs/rGO lactate sensor reported by Phamonpon et al. [60].

Different electrochemical techniques offer unique advantages for lactate biosensors. DPV, AMP, and EIS each provide a high sensitivity and specificity, making them suitable for various applications. The choice of technique depends on the specific requirements of the sensor, such as the desired detection limit, response time, and application context. Future research should focus on integrating these techniques to develop more robust and versatile lactate biosensors. The examples presented highlight the significant potential for technological innovation and application development in lactate electrochemical biosensors. Continuous advancements in sensor design, novel materials, and integration technologies are expected to enhance the precision and speed of lactate detection, positioning these sensors as crucial tools in disease diagnosis, health monitoring, and biomedical research across diverse fields. Here are some of the major types and applications of lactate electrochemical biosensors based on different detection methods.

## 6. Implantable and Non-Implantable Miniaturized Lactic Acid Electrochemical Biosensors

Implantable and non-implantable miniaturized electrochemical biosensors provide two key options for electrochemical sensors. Implantable sensors are designed for in vivo monitoring and provide the real-time, accurate monitoring of lactate concentrations directly in tissues or organs, making them important in clinical settings and critical care environments [93]. Non-implantable biosensors, on the other hand, provide the non-invasive monitoring of lactate levels in accessible biofluids. This can make them suitable for applications such as sports medicine and chronic disease management [88]. These two detection solutions effectively meet the need for high-sensitivity and real-time monitoring, while also accommodating the demand for flexible, wearable devices.

### 6.1. Implantable Electrochemical Lactate Biosensors

Implantable electrochemical lactate biosensors are designed for continuous in vivo monitoring. These sensors are implanted into tissues or organs, providing real-time lactate data, which are crucial for patients needing intensive monitoring. Biocompatible materials are used to minimize immune reactions and tissue damage. Omar et al. developed a biodegradable and biocompatible implantable lactate sensor featuring a magnesium electrode on a flexible PLA substrate, designed for seamless integration into cardiac tissue. The sensor demonstrated an excellent lactate detection performance with an enhanced sensitivity of 1.11 µA mM^−1^ and a high stability. It fully degrades within 24 h post-implantation without requiring surgical removal, making it highly suitable for continuous cardiac monitoring and other implantable sensor applications [94]. This will help to assess metabolic states and adjust treatment plans in real time. Miniaturization enhances patient comfort and enables long-term monitoring.

Recent advancements in sensor technology have led to the development of implantable sensors with wireless data transmission capabilities, significantly enhancing their usability. Among these, microneedle sensors stand out for their ability to minimally penetrate the skin to detect the lactate levels in blood or interstitial fluid [70]. Tehrani et al. developed a fully integrated wearable microneedle array sensor for the continuous monitoring of lactate in interstitial fluid, utilizing amperometry and chronoamperometry (This summary chart is detailed in Figure 6A at the end of this section) [95]. This microneedle sensor can painlessly penetrate the epidermis, providing direct ISF access and a stable performance for up to 12 h, demonstrating a strong correlation (Pearson’s r = 0.94) with blood lactate levels during daily activities. To achieve a high selectivity in blood, Tehrani et al. utilized a functionalized gold electrode surface and applied an enzyme-based method to immobilize LOx on the sensor. Gold electrodes are well-known for their excellent electrical conductivity, while LOx specifically catalyzes the oxidation of lactate. The combination of these two components ensures both the conductivity and selectivity of the sensor, thereby minimizing interference from the simultaneous presence of other metabolites such as glucose during detection. This approach effectively minimizes discomfort and reduces the risk of infection, establishing microneedle sensors as a highly promising tool for continuous, non-invasive monitoring [96,97,98,99]. Zhong et al. developed a wearable microneedle biosensor array with 1200 µm long microneedles and applied it to the multiplexed detection of biomarkers such as glucose, lactate, and alcohol in interstitial fluid. This design provides a nearly painless, minimally invasive method for continuous monitoring with a lactate sensitivity of 150 nA mM^−1^ cm^−2^ and a detection limit of 0.5 mM [100] (This summary chart is detailed in Figure 6B at the end of this section). Dai et al. developed a wearable sensor patch with 600 µm hydrogel microneedles for the in situ detection of lactic acid in interstitial fluids, achieving a sensitivity of 3 ± 0.4 nA mM^−1^ and a linear range of 0.1–12 mM [101] (This summary chart is detailed in Figure 6C at the end of this section).

### 6.2. Non-Implantable and Miniaturized Electrochemical Lactate Biosensors

Non-implantable electrochemical lactate biosensors are designed for the external monitoring of lactate levels in biological fluids such as sweat, saliva, or tears, offering a non-invasive approach suitable for various applications. These sensors are widely used in sports medicine to track athlete performance, providing real-time lactate data without the need for invasive sampling and enhancing training and recovery management.

He et al. developed a textile-based non-invasive electrochemical sensor to monitor the lactic acid in sweat using a nitrogen-doped carbon textile (SilkNCT) derived from silk fabric. The unique woven structure of SilkNCT enhances electrical conductivity and sweat uptake, enabling effective real-time biomarker analysis [102] (This summary chart is detailed in Figure 6D at the end of this section). Rabost-Garcia et al. developed a non-invasive multiparametric approach using sweat lactate sensors combined with sweat rate and heart rate measurements, applying neural network algorithms to accurately predict blood lactate levels [103]. This method significantly improved the equivalence of sweat-based lactate monitoring to blood lactate, reducing the accumulated error to just 0.3 mM compared to standard blood lactate meters.

These miniaturized and portable sensors are also crucial in clinical diagnostics for metabolic disorders, offering easy-to-use, portable solutions for home health monitoring. Recent advancements have integrated these sensors into compact and portable devices, such as small fitness trackers and smartwatches, enabling the continuous monitoring of health conditions during everyday life. Their flexible, miniaturized, and portable design further enhances their wearability, comfort, and user compliance, making them particularly suitable for long-term and convenient health monitoring. Saha et al. developed a miniaturized, portable, wearable lactate sensor by integrating osmotic hydrogels and microfluidic channels for continuous sweat extraction and detection. This compact sensor, designed for long-term use, can achieve a high sensitivity (117 nA mM^−1^ mm^−2^) with an ultra-low power consumption, with a detection limit of 350 nM, making it ideal for real-time monitoring during various physical activities [104]. Zhu et al. developed a miniaturized, portable electrochemical biosensor utilizing a functionalized poly(3,4-ethylenedioxythiophene) (PEDOT) film for non-invasive lactate detection in sweat [68]. This compact sensor, with a flexible and thin design, enables the continuous real-time monitoring of lactate levels for up to 8 h, with a less than 5.5% error during prolonged use, demonstrating its excellent stability, portability, and suitability for wearable applications.

Non-implantable miniaturized sensors also utilize hydrogel-based designs for detecting lactate in tear fluid [104], sweat [105], or saliva [53], providing versatile options for non-invasive health assessment. Integrating microfluidics with these miniaturized sensors allows for the precise control of sample flow and reaction conditions, replicating different monitoring scenarios and enabling continuous lactate measurement in complex biological samples [106,107]. This miniaturized and integrated approach demonstrates significant potential for high-throughput, automated lactate detection, contributing to personalized healthcare and performance optimization.

**Figure 6 sensors-25-01045-f006:**
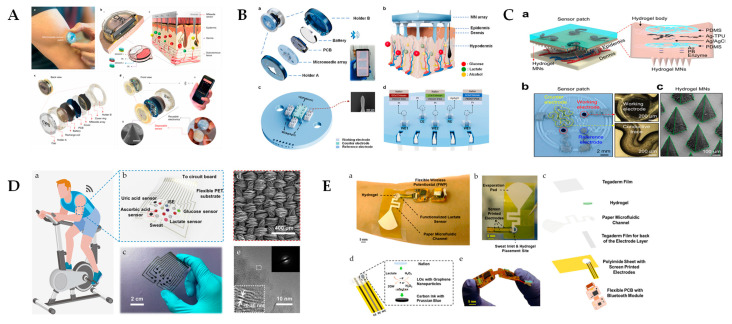
(**A**) Schematic illustration of the multiplexed microneedle-based wearable sensor system and its subcomponents reported by Tehrani et al. [95]; (**B**) overall design of the multiplexed microneedle-based wearable sensor system reported by Zhong et al. [100]; (**C**) schematics and images of the sensor patch reported by Dai et al. [101]; (**D**) wearable sweat analysis patch based on SilkNCT reported by He et al. [102]; and (**E**) osmotic wearable for lactate sensing in sweat for continuous biochemical monitoring reported by Saha et al. [105].

## 7. Electrochemical Lactate Biosensors Based on Multiple Advanced Technologies

In addition, there are many different technologies (organic electrochemical transistors (OECTs), nanozymes, and machine learning) driving the rapid development of lactate electrochemical biosensors.

OECTs represent a novel class of electrochemical sensors that leverage organic semiconductor materials to provide a high sensitivity and low power consumption. These transistors operate by using the organic semiconducting channel (such as PEDOT) to detect lactate ions through an electrochemical process, amplifying the electrical signal generated by the lactate concentration (This summary chart is detailed in Figure 7A at the end of this section). The unique advantage of OECTs lies in their ability to work at low voltages, which minimizes power consumption and reduces the need for complex instrumentation. A detailed schematic of OECTs and their working principle could help to elucidate how they amplify signals in lactate detection. Moreover, a comparison with potentiometric sensors, which measure the potential differences caused by ion concentrations, would highlight the increased sensitivity and stability offered by OECTs. Gualandi et al. developed an organic electrochemical transistor-based lactate sensor utilizing a PEDOT channel with a gate electrode modified by Ni/Al layered double hydroxide and lactate oxidase. The transistor architecture amplifies the signal, enhancing sensitivity and lowering the detection limit for wearable applications [108]. The sensor utilizes PEDOT as its conductive organic material, which provides a good conductivity and ensures that the sensor operates at low voltages while maintaining a high sensitivity. Ni/Al layered double hydroxides (LDH) are commonly used as gate electrode modifiers, and their layered structure provides a larger surface area and ion-exchange capacity, which enhances the sensor’s response to lactate, improves the electrochemical performance of the OECT, and enhances the current amplification effect.

Nanozymes are nanomaterials with enzyme-like activity that catalyze lactate oxidation, enhancing sensor performance. Compared to traditional enzymes, nanozymes offer a higher stability and longer service life. Incorporating nanozymes into lactate biosensors improves their sensitivity and reliability in complex biological samples. Komkova et al. developed a nanozyme-enhanced lactate biosensor using Prussian Blue nanoparticles and lactate oxidase immobilized within a siloxane-perfluoro sulfonated ionomer composite membrane. The integration of Prussian Blue nanozymes increased the sensitivity to 4.4 ± 0.5 µA mM^−1^ cm^−2^ and allowed for continuous operation for several hours at high lactate concentrations (up to 80 mM) while maintaining 90% of its initial activity, demonstrating suitability for real-time, wearable applications [109] (This summary chart is detailed in Figure 7B at the end of this section). These nanomaterials significantly improve detection limits and response speed.

In addition, machine learning technology has introduced new methods for data processing and analysis in electrochemical biosensors. Machine learning algorithms enable the real-time analysis and prediction of sensor data, identifying lactate concentration trends. Jingfeng et al. developed a gold lactate sensor integrated with machine learning to enhance drug susceptibility testing in patient-derived tumor models [110]. Machine learning techniques enable the evaluation of multiple metrics during electrochemical testing and can achieve correlation coefficients as high as 0.9. This approach allows for the dynamic, repeated monitoring of samples, significantly reducing the risk of misdiagnosis. This enhances sensor detection accuracy and supports personalized health management. It also reduces the impact of environmental factors on detection results, thus improving sensor reliability [103]. Zhou et al. developed a machine learning-enhanced nonenzymatic electrochemical biosensor for lactate and glucose detection, utilizing a microdroplet array integrated with a backpropagation neural network. The incorporation of the neural network model significantly improved the selectivity of the sensor, enabling the accurate detection of glucose and lactate concentrations in multi-analyte mixtures (0.25–20 mM) with a correlation coefficient of 0.9997 [111] (This summary chart is detailed in Figure 7C at the end of this section).

These technologies also expand the application prospects of lactate sensors in medical, sports, and health monitoring. By integrating various fields, lactate biosensors become more efficient, reliable, and versatile. This multidisciplinary approach ensures continuous innovation and improvement in lactate monitoring solutions.

**Figure 7 sensors-25-01045-f007:**
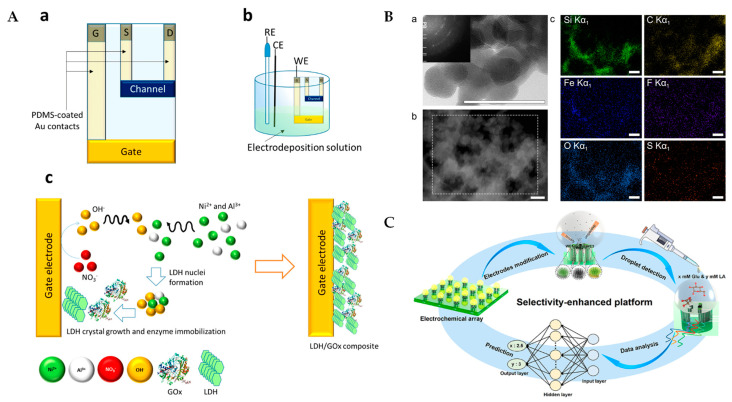
(**A**) Schematic diagram of OECT sensor. Scheme of the device geometry (**a**), experimental setup employed for the gate electrode modification (**b**), processes occurring at the gate electrode, where nitrate reduction induces an increase in OH^−^ concentration, resulting in the precipitation of the LDH/enzyme composite (**c**) [108]; (**B**) characterization of core materials for nanozyme–enzyme co-immobilized sensors. (**a**) High-resolution transmission electron microscopy image of catalytically synthesized Prussian Blue nanozyme. Inset: selected area electron diffraction pattern. (**b**) High-angle annular dark-field scanning transmission electron microscopy (HAADF-STEM) of the γ-aminopropyltriethoxysilane-perfluorosulfonated ionomer membrane, containing lactate oxidase, Prussian Blue-based nanozyme and carbon black nanoparticles, and (**c**) energy-dispersive X-ray spectroscopy maps for the same spot, indicating distribution of elements in the sensing layer [109]; and (**C**) illustration for achieving highly specific nonenzymatic electrochemical sensing by the neural network [111].

## 8. Conclusions

This review provides a comprehensive overview of the recent advancements, applications, and challenges of lactate electrochemical biosensors in the medical field. As lactate is a critical biomarker for assessing metabolic status and clinical conditions, the development of these biosensors highlights their significant potential for rapid and accurate lactate monitoring in healthcare settings. Despite their numerous advantages, real-world applications still face several obstacles, such as the complexity of biological samples, variations in environmental conditions like temperature and pH, and interference from substances commonly found in critically ill patients. To offer a comprehensive overview of the various strategies and materials used in lactate electrochemical sensors, Table 4 summarizes the key articles reviewed in this study, outlining their sensor design strategies, detection methods, and relevant performance metrics. This table serves as a valuable reference for understanding the progress in the field, emphasizing the diversity of approaches and the ongoing challenges in developing reliable, efficient, and adaptable lactate sensors for clinical use. By comparing these studies, we aim to provide a clearer understanding of the advancements in sensor technology and identify areas for future improvement.

Future research should focus on optimizing sensor design, enhancing the stability of biorecognition elements, and improving resistance to interference through advanced signal processing and enhancement technologies. Additionally, expanding both implantable and non-implantable sensor options will offer a greater flexibility and convenience, catering to diverse clinical needs. Through continued technological innovation and interdisciplinary collaboration, lactate electrochemical biosensors are expected to advance personalized medicine and continuous health monitoring. At the same time, these sensors will provide valuable support to clinicians and researchers in disease diagnosis, treatment planning, and patient management.

## Figures and Tables

**Figure 1 sensors-25-01045-f001:**
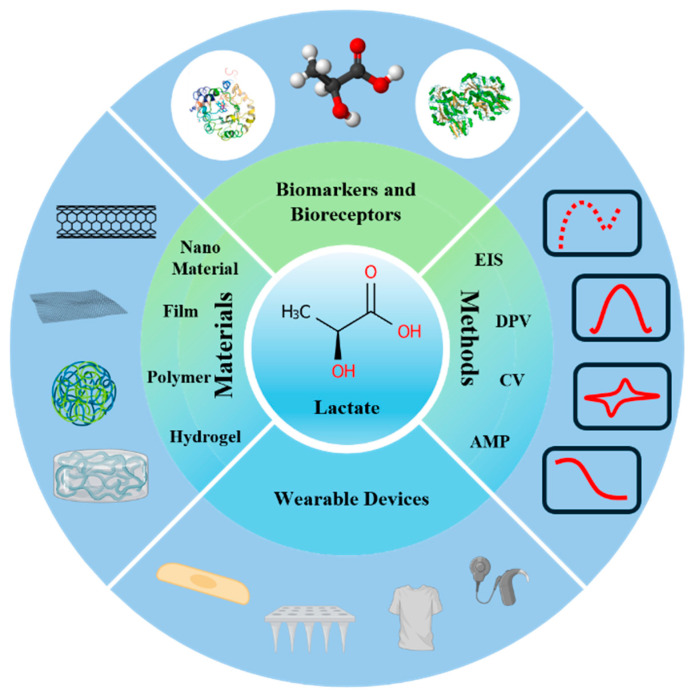
Lactate sensors are used for clinical detection.

**Figure 2 sensors-25-01045-f002:**
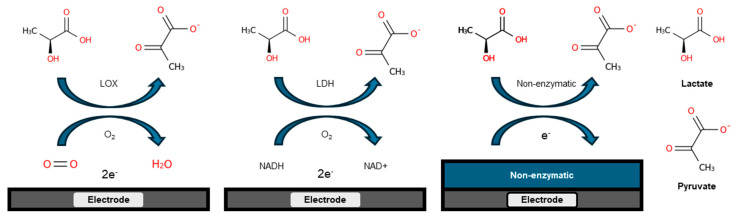
Schematics illustrating the detection principle of enzyme/non-enzyme lactate sensors.

**Figure 3 sensors-25-01045-f003:**
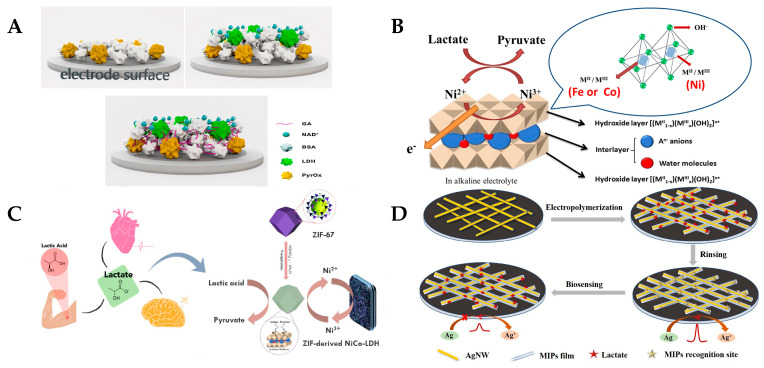
(**A**) Schematic diagram of LDH-NAD+/PyrOx biosensor centered on lactate oxidase [49]; (**B**) schematic illustration of lactate oxidation on the dual active site of bimetallic Ni-based LDHs with various transition metals [44]; (**C**) schematic illustration of ZIF-67-derived NiCo LDH for lactate detection by wearable biosensors [45]; and (**D**) schematic preparation of MIPs-AgNW electrochemical biosensors for epidermal monitoring of lactate [46].

**Table 1 sensors-25-01045-t001:** Lactate levels in physiological and pathological conditions.

Physiological State/Condition	Concentration Range (mM)	Description	Ref.
Normal Metabolism (Arterial)	0.5–1.5	Baseline arterial lactate levels in resting healthy individuals.	[2]
Mild Activity/Stress State	2–4	Elevated during moderate exercise or mild metabolic stress.	[7]
Intense Activity/Vigorous Exercise	15–20	Muscle lactate accumulates locally, higher than blood levels.	[8]
Tumor Microenvironment	10–30	Tumor lactate levels elevated, promoting progression and resistance.	[29]
ICU Patients (Arterial)	≥2	Diagnostic for septic shock; correlates with severity.	[12]
Critically Ill Patients (Lactic Acidosis)	≥4	Elevated lactate in shock or severe metabolic derangements.	[15]
Trauma Patients	≥4	Indicates inadequate perfusion, associated with high mortality.	[22]
Critical Threshold for Lactate (Emergency)	≥8	Indicates critical state; high 30-day mortality risk.	[23]

**Table 2 sensors-25-01045-t002:** Summary of Recent Reviews on Lactate Sensors: Contributions and Limitations.

Theme	Year	Main Contributions	Limitations	Ref.
Electrochemical lactate detection	2019	Reviewed enzyme electrochemical methods and explored applications in critical care.	Discussion of non-enzymatic sensors is lacking.	[30]
Translational lactate sensing	2021	Explored the status of lactate biosensors for non-invasive real-time monitoring in the medical field.	Lacks in-depth analysis of core detection methods for lactate sensors.	[31]
Multi-fluid lactate sensing	2021	An overview of electrochemical sensors for the detection of lactate in a wide range of human fluids.	Lack of in-depth exploration of materials chemistry.	[32]
Enzymatic vs. non-enzymatic sensors	2021	Reviewed enzymatic and non-enzymatic sensors, focusing on their modifying materials and immobilization techniques.	A description of the medical field needs to be added.	[33]
Real-life lactate monitoring	2022	Sensor applications in the agri-food and clinical fields were reviewed, respectively.	Attention to innovative materials and emerging technologies needs to be increased.	[34]
Nanomaterial advancements	2023	Focused on non-enzymatic lactate sensors constructed from various nanomaterials.	Lack of discussion on specific feasibility for clinical and real-life applications.	[35]
Non-invasive lactate monitoring	2024	Described lactate sensors (based on sweat and interstitial fluids) for non-invasive detection in critical care.	No focus on cross-application of sensors with emerging technologies.	[36]

**Table 3 sensors-25-01045-t003:** Electrochemical biosensor for lactate.

Material	Examples	Advantages	Disadvantages	Ref.
AuNPs	Nanomaterials	High surface area, high conductivity, excellent catalytic activity	Cost of synthesis	[66]
CNTs	Nanomaterials	Excellent electrocatalytic performance, enhanced electron transfer	Stability issues, complex fabrication	[62]
Graphene	Nanomaterials	High surface area, good electrical conductivity	Stability issues, high cost, easily oxidized	[63]
MOFs	Nanomaterials	High surface area, tunable structure	Synthesis complexity, complex fabrication	[64]
MXenes	Nanomaterials	High conductivity, large surface area	Complex synthesis	[65]
PC nanoporous membranes	Membrane Materials	Selective permeability, enhanced sensitivity	May not block all interfering substances	[69]
Nafion	Membrane Materials	Selective permeability, protects enzyme activity	Limited permeability for small molecules, low conductivity	[70]
PPy	Polymer Materials	Good conductivity, enhanced electrochemical activity	Mechanical strength may degrade	[71]
PANI	Polymer Materials	Good conductivity, tunable properties	Cost of synthesis	[72]
PVA	Polymer Materials	Good biocompatibility, enhanced enzyme stability	Mechanical stability issues	[74]
PAM hydrogels	Hydrogel Materials	High water content, biocompatible	Loss of water content, may deform	[76]
SA hydrogels	Hydrogel Materials	High biocompatibility, tunable structure	Easily degradable, weak conductivity	[73]

**Table 4 sensors-25-01045-t004:** Electrochemical biosensors for lactate.

Electrode	Targets	Methods	Detection Range	LOD	Sensitivity	Imp.^1^	Ref.
PC/LOx/CS	Lac, Temp	AMP	10 µM–35 mM	144 nM	82.4 nA mM^−1^	N	[69]
MIPs-AgNWs	Lac	DPV	1 µM–100 mM	220 nM	4.5 µA dacade^−1^	N	[46]
PEDOT:PSS/CS/LOx	Lac, Glu, Alc	AMP	500 µM–5 mM	250 nM	150 nA mM^−1^ cm^−2^	I	[100]
Hydrogel/LOx	Lac	AMP	0 mM–15 mM	350 nM	9 µA mM^−1^ mm^−2^	N	[105]
AuE/rGO-AgNPs/MIP	Lac	CV	10 µM–250 µM	726 nM	0.12428 µA mM^−1^ −0.14102 µA mM^−1^	O	[86]
PB/GO/Au/LOx	Lac	AMP	1 µM–222 µM 222 µM–25 mM	800 nM	40.6 µA mM^−1^ cm^−2^ 1.9 µA mM^−1^ cm^−2^	N	[48]
Rgo/PtNPs/LOx	Lac	AMP	0 mM–10 mM	2.04 µM	43.96 µA mM^−1^ cm^−2^	I	[70]
PEDOT:PSS/TTF/CS/Ldh	Lac, pH	AMP	140 µM–13.32 mM	5 µM	1.1 µA dacade^−1^	I	[94]
LOx-GO-ZnO	Lac	AMP	15 µM–1.25 mM	9 µM	3.308 µA mM^−1^	N	[53]
LOx/BzA/GNR	Lac	CV	34 µM–280 µM	11 µM	5.5 µA mM^−1^	O	[87]
PVC/CHI- LOx/PB/C-MN	Lac	AMP	250 µM–35 mM	14.8 µM	−8.04 nA mM^−1^	I	[96]
LDH-NAD+/PyrOx/SPE	Lac	EIS	10 µM–250 µM	17 µM	62.7 Ω cm^2^ mM^−1^	O	[49]
LOx/PmPD/IrO_2_/PBE	Lac, pH	AMP	0 mM–3 mM	19 ± 7 µM	2.63 ± 0.66 nA mM^−1^	I	[80]
LOx/PB enzyme-nanozyme	Lac	AMP	20 µM–100 mM	20 µM	4.4 ± 0.5 µA mM^−1^ cm^−2^	N	[109]
SWCNT/NiCo_2_O_4_/HRP/LOx	Lac	AMP	100 µM–30 mM	39.9 µM	98 nA mM^−1^	N	[62]
PtNPs/rGO/LOx	Lac	AMP	0 mM–25 mM	70 µM	0.87 µA mM^−1^ cm^−2^	N	[60]
LOx/CS/PEDOT	Lac	DPV	250 µM–1 mM 1 mM–40 mM	83 µM	43.42 µA mM^−1^ cm^−2^ 0.32 µA mM^−1^ cm^−2^	N	[68]
PANI	Lac, pH	AMP	250 µM–10 mM 10 mM–60 mM	83 µM	18.62 nA mM^−1^ 4.25 nA mM^−1^	N	[112]
LOx/CS/Pt	Lac	AMP	200 µM–3 mM	110 µM	35.8 µA mM^−1^ cm^−2^	O	[113]
SilkNCT/PtNPs/LOx	Lac, Glu, AA, UA, Na+, K+	AMP	5 mM–35 mM	500 µM	174 nA mM^−1^	N	[102]
PA/GO/PANHS/LOx	Lac	EIS	1.3 mM–113.4 mM	1 mM	N/A	N	[77]
NDG/LOx/PMM70	Lac, Glu	AMP	0 mM–25 mM	6.5 mM	443.2 nA mM^−1^	O	[76]
Hydrogel MNs/GA/BSA/LOx	Lac, Glu	AMP	100 µM–12 mM	N/A	3 ± 0.4 nA mM^−1^	I	[101]
LOx/PB/Au/PS	Lac, ATP	AMP	1 µM–10 mM	N/A	101 nA mM^−1^	O	[79]

^1^ Imp. represents implantable sensors, ‘I’ represents implantable sensors, ‘N’ represents non-implantable sensors, and ‘O’ represents other sensors.

## Data Availability

Not applicable. This review article does not contain raw data; all analyzed information is available from the cited references.

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
