# Peer review of "A Comprehensive Review of Advanced Lactate Biosensor Materials, Methods, and Applications in Modern Healthcare"

_sensors, 2025, doi:10.3390/s25041045_

Round 1

Reviewer 1 Report

Comments and Suggestions for Authors

 1. The authors primarily present schematics illustrating the detection principles of lactate sensors; however, Figure 2 is limited to enzymatic and electrochemical sensors, which could mislead the reader about lactate sensor technologies. You shall add similar conceptual figure for supporting 3.2. Enzyme-Free . Notably, major reviews on lactate sensors have already been published, but these references are not cited in this manuscript. It is recommended that the authors compile a table summarizing existing reviews or book chapters on lactate sensors, providing a brief overview of their content. Many reviews are missing. This will help the reader understand the current landscape and also highlight the unique contributions of this review, distinguishing it from previous publications. 

2. The information presented in Table 1, "Electrochemical Biosensor for Lactate," is currently too limited and requires expansion. The authors should include additional original research papers to provide a more comprehensive overview of the field. Additionally, the term "PH" appears to be incorrectly used and should be corrected for accuracy. To improve clarity and accessibility, I recommend reordering the rows of the table by the detection range, from low to high, based on the lower limit of the reported detection range for each sensor. This reorganization will help readers better compare the performance of different biosensors.

3. The figures presented in the review are too small, and the text size within many of them is particularly difficult to read. This limits the clarity and overall impact of the figures. I recommend increasing the size of both the figures and the text within them to ensure that they are legible and effectively convey the necessary information.

4. The statement regarding lactate levels in the bloodstream, including the normal range and elevated levels under specific conditions (referring to… the normal level of lactate in the bloodstream of a human is usually between 0.5-2.2 mmol/L, but in situations such as exercise, shock, severe trauma, or tissue damage, lactate levels can rise to 5-15 mmol/L…..), would benefit from a more comprehensive citation of biomedical and medical literature. To improve the reliability of this information, I suggest referencing studies from reputable biomedical journals and summarizing the lactate concentrations across various conditions in a well-organized table. This would provide readers with a clearer, evidence-based overview of lactate levels in different physiological states, enhancing the overall accuracy and depth of the review.

5. The organization of the manuscript is somewhat confusing, particularly in the use of the term "membrane." Generally, "membrane" should refer to a selective barrier sheet, which allows specific molecules, such as lactate, to pass through. However, some examples presented may be more accurately described as films rather than membranes. To enhance clarity, I recommend the authors carefully distinguish between membranes and films, and provide detailed explanations of the surface chemistry, material properties, and mechanisms behind selective barrier membranes. Additionally, since some membranes are polymer-based, this overlap may lead to confusion. A more structured approach to grouping materials based on their properties and functions would help clarify these concepts and improve the overall flow of the review.

6. The manuscript just re-reports existing work without offering enough a critical analysis or comparison of the different methods and materials discussed. To enhance the value of the review, I recommend the authors provide a more in-depth critique of the studies, highlighting the strengths and weaknesses of each method and material. A comparison of the various approaches, along with a discussion of their respective pros and cons, would provide readers with valuable insights and a clearer understanding of the current state of the field. This would also help position the review as a more analytical and comprehensive resource.

7. The section divisions in the manuscript are unclear and could benefit from further refinement. For example, the section on "Non-Implantable Electrochemical Lactate Biosensors" is too broad?? To improve focus, this section may be narrowed to provide a more specific discussion of the technologies within that category. Additionally, it would be valuable to explore miniaturized, portable devices for lactate detection, particularly those that are not worn on the body. Expanding on this topic would help cover a wider range of devices and enhance the comprehensiveness of the review.

8. The authors should consider adding a figure to illustrate organic electrochemical transistors (OECTs) and provide a basic explanation of their operating principles. Additionally, a comparison with the potentiometric sensor principle would be valuable to highlight the advantages and differences between these technologies. It is also important for the authors to describe the insights into the principles and materials used in the adaptation of OECTs for lactate detection.

9.  The discussion on sensitivity in the manuscript is unclear and lacks consistency, particularly with the use of different units, such as ….. A mM cm⁻², and the lack of normalization by sensor area. To improve clarity and informativeness, the authors should provide a more detailed comparison of sensitivity values, ensuring they are consistently presented with proper normalization. A discussion of which materials provide higher response, based on normalized sensitivity, would help readers better understand the performance of different materials in lactate biosensing.

Comments on the Quality of English Language

The manuscript contains numerous issues with English structure and typographical errors, which detract from its overall quality. For example, in the heading "Advanced Technologie," the plural form "Technologies" should be used, and an apostrophe is needed in some instances. I recommend the authors carefully proofread the entire manuscript to correct these errors and improve the language quality.

Author Response

Reviewers' comments:

  1. The authors primarily present schematics illustrating the detection principles of lactate sensors; however, Figure 2 is limited to enzymatic and electrochemical sensors, which could mislead the reader about lactate sensor technologies. You shall add similar conceptual figure for supporting 3.2. Enzyme-Free. Notably, major reviews on lactate sensors have already been published, but these references are not cited in this manuscript. It is recommended that the authors compile a table summarizing existing reviews or book chapters on lactate sensors, providing a brief overview of their content. Many reviews are missing. This will help the reader understand the current landscape and also highlight the unique contributions of this review, distinguishing it from previous publications.

Response 1: Thank you for pointing this out. In response to your comments, we have made the necessary corrections to the manuscript. A conceptual diagram for enzyme-free sensors has been added to Figure 2 (line #150), and the detection principles of enzyme-free lactate sensors are explained in the following section (lines #191-196), as described in Section 3.2. This new figure complements the existing schematic diagrams of enzyme-based and electrochemical sensors, providing a more comprehensive view of the various lactate sensor technologies. Additionally, we have compiled Table 2 (line #102), summarizing existing reviews on lactate sensors and briefly outlining their main contributions. This will help readers gain a better understanding of the current state of lactate sensor technologies and highlight the unique contributions of our review, distinguishing it from previous publications in the field (lines #91-100).

  1. The information presented in Table 1, "Electrochemical Biosensor for Lactate," is currently too limited and requires expansion. The authors should include additional original research papers to provide a more comprehensive overview of the field. Additionally, the term "PH" appears to be incorrectly used and should be corrected for accuracy. To improve clarity and accessibility, I recommend reordering the rows of the table by the detection range, from low to high, based on the lower limit of the reported detection range for each sensor. This reorganization will help readers better compare the performance of different biosensors.

Response 2: Thank you for your recognition and thorough review of our work. In response to your suggestions, we have expanded Table 1 by including additional original research papers, providing a more comprehensive overview of electrochemical lactate biosensors (line #630). We also apologize for our oversight regarding the use of "PH" in the table. We have corrected it to "pH" (line #150) to improve accuracy. To enhance clarity and comparability for the readers, we have reorganized the rows of Table 1 based on the lower limit of the reported detection range for each sensor, ordered from low to high. Lastly, the revised Table 4(line #630) has been moved to follow Section 7  to improve the logical flow of the manuscript.

  1. The figures presented in the review are too small, and the text size within many of them is particularly difficult to read. This limits the clarity and overall impact of the figures. I recommend increasing the size of both the figures and the text within them to ensure that they are legible and effectively convey the necessary information.

Response 3: Thank you for your valuable suggestion. In response, we have increased the size of all figures and the text within them (lines #53, 88, 102, 150, 235, 319, 354, 437, 552, 616, 630) to match the font size used in the main text of the manuscript. This ensures improved clarity and overall impact, allowing readers to effectively interpret the figures. Additionally, we have enlarged and bolded the labels for each figure (A, B, C, etc.) to make them more distinguishable, ensuring that readers can clearly identify the corresponding sections of the figures during their review.

  1. The statement regarding lactate levels in the bloodstream, including the normal range and elevated levels under specific conditions (referring to… the normal level of lactate in the bloodstream of a human is usually between 0.5-2.2 mmol/L, but in situations such as exercise, shock, severe trauma, or tissue damage, lactate levels can rise to 5-15 mmol/L…..), would benefit from a more comprehensive citation of biomedical and medical literature. To improve the reliability of this information, I suggest referencing studies from reputable biomedical journals and summarizing the lactate concentrations across various conditions in a well-organized table. This would provide readers with a clearer, evidence-based overview of lactate levels in different physiological states, enhancing the overall accuracy and depth of the review.

Response 4: Thank you for pointing this out. Your recommendation is highly insightful, and in response, we have incorporated additional references from reputable biomedical journals in the Introduction to provide a more comprehensive citation of lactate levels in the bloodstream under various physiological conditions. We have referenced the classification of lactate concentrations from the studies and have organized the lactate concentration ranges into 0.5-1.5 mmol/L (line #42), 2-4 mmol/L (line #44), and 15-20 mmol/L (line #46), based on different conditions. Furthermore, we not only discuss the variation in lactate concentrations under various circumstances (lines #44-47) but also address the use of lactate concentration thresholds in medical practice to assess disease severity (lines #55-87), which will help readers better understand the lactate levels in different contexts. Additionally, we have summarized the lactate concentrations under these conditions in a well-organized table (line #88), which will assist readers in clearly visualizing the fluctuations in lactate levels, thereby enhancing the overall accuracy and depth of the review.

  1. The organization of the manuscript is somewhat confusing, particularly in the use of the term "membrane." Generally, "membrane" should refer to a selective barrier sheet, which allows specific molecules, such as lactate, to pass through. However, some examples presented may be more accurately described as films rather than membranes. To enhance clarity, I recommend the authors carefully distinguish between membranes and films, and provide detailed explanations of the surface chemistry, material properties, and mechanisms behind selective barrier membranes. Additionally, since some membranes are polymer-based, this overlap may lead to confusion. A more structured approach to grouping materials based on their properties and functions would help clarify these concepts and improve the overall flow of the review.

Response 5: Thank you for pointing this out. We have addressed your comment by adding a clear definition of "membrane" in the section "Membrane-Based Lactate Electrochemical Biosensors" (line #269), which will help readers better understand the specific role of membranes as selective barrier sheets. To enhance clarity, we have also provided examples of different selective membrane materials (lines #275-278, 284-287) and explained the surface chemistry, material properties, and mechanisms behind selective barrier membranes (lines #280-283, 288-291). Additionally, we have reorganized the discussion, grouping the materials mentioned in Section 4 based on their properties and functions, and summarized the advantages and disadvantages of different modified materials in Table 3 (line #355). Additionally, in the table 3, the polymer-based PC membrane has been specifically named as "PC nanoporous membranes," which helps readers distinguish these materials and clarifies the concepts, improving the overall flow of the review.

  1. The manuscript just re-reports existing work without offering enough a critical analysis or comparison of the different methods and materials discussed. To enhance the value of the review, I recommend the authors provide a more in-depth critique of the studies, highlighting the strengths and weaknesses of each method and material. A comparison of the various approaches, along with a discussion of their respective pros and cons, would provide readers with valuable insights and a clearer understanding of the current state of the field. This would also help position the review as a more analytical and comprehensive resource.

Response 6: Thank you for pointing this out. To enhance the value of this review, we have added a critical analysis at the end of Section 4 (line #324-353), where we compare the different materials used in electrochemical lactate sensors. This analysis highlights the strengths and weaknesses of each material, providing readers with a deeper understanding of how various materials impact sensor performance and their respective applications. We have categorized these materials based on their properties and created Table 3 (line #354). Additionally, the relevant characteristics of sensors based on different materials have been incorporated into Table 4 (line #630), where the detection limits of different sensors are arranged from low to high. This will help readers better understand the current applications of each material. These additions have increased the value of the review and will provide valuable insights for the readers.

  1. The section divisions in the manuscript are unclear and could benefit from further refinement. For example, the section on "Non-Implantable Electrochemical Lactate Biosensors" is too broad?? To improve focus, this section may be narrowed to provide a more specific discussion of the technologies within that category. Additionally, it would be valuable to explore miniaturized, portable devices for lactate detection, particularly those that are not worn on the body. Expanding on this topic would help cover a wider range of devices and enhance the comprehensiveness of the review.

Response 7: Thank you for pointing this out. To provide a more focused review of "Non-Implantable Electrochemical Lactate Biosensors" (line #512), we have narrowed the scope of the section and provided a more targeted discussion. The revised section now emphasizes non-invasive, portable, and miniaturized devices for lactate detection. This includes highlights the importance of miniaturization (lines #533-543). This adjustment allows for a more specific and in-depth exploration of these technologies, expanding the coverage to include a wider range of devices, thereby enhancing the comprehensiveness of the review.

  1. The authors should consider adding a figure to illustrate organic electrochemical transistors (OECTs) and provide a basic explanation of their operating principles. Additionally, a comparison with the potentiometric sensor principle would be valuable to highlight the advantages and differences between these technologies. It is also important for the authors to describe the insights into the principles and materials used in the adaptation of OECTs for lactate detection.

Response 8: Thank you for pointing this out. We realized that we overlooked the description of emerging technologies, so we have added a conceptual description of organic electrochemical transistors (OECTs) (line #570-574), including an explanation of their operating principles and how they are applied in lactate detection. Additionally, we compared the operating principles of OECTs with those of traditional three-electrode electrochemical sensors to help readers better understand OECT sensors (line #583-589). To further enhance understanding, we have also included a schematic diagram (Fig. 7A) to illustrate the principles of OECTs (line #622). Furthermore, we have supplemented diagrams of Nanozyme (Fig. 7B) (line #622) and machine learning technology (Fig. 7C) (line #622). These additions help readers better understand how emerging technologies, such as OECTs, can be integrated with traditional electrochemical sensors for lactate detection.

  1. The discussion on sensitivity in the manuscript is unclear and lacks consistency, particularly with the use of different units, such as ….. A mM cm⁻², and the lack of normalization by sensor area. To improve clarity and informativeness, the authors should provide a more detailed comparison of sensitivity values, ensuring they are consistently presented with proper normalization. A discussion of which materials provide higher response, based on normalized sensitivity, would help readers better understand the performance of different materials in lactate biosensing.

Response 9: Thank you for pointing this out. We have made efforts to standardize the sensitivity units for various sensors to μ(n)A mM cm⁻² and have considered sensor area whenever possible. However, some studies did not provide sensor area information (lines #277-283, 486, et al.), and in cases where sensors constructed using the EIS (electrochemical impedance spectroscopy) method were involved, sensitivity was reported in Ω cm² mM⁻¹. To make the sensor performance more clearly understandable to readers, we have reorganized the manuscript as per your second suggestion, using the detection limit (LOD) as the primary reference for comparing sensor performance. The performance indicators of the sensors are now listed in Table 4(lines #630), with sensors arranged in ascending order of the lower detection limit as the criterion for performance evaluation. This approach will help readers better understand the performance of sensors with different materials in lactate detection.

Reviewer 2 Report

Comments and Suggestions for Authors

In consideration of the pivotal function of lactate in cell metabolism, precise real-time monitoring of this substance is imperative. The article provided for review offers an exhaustive examination of electrochemical biosensors that are pertinent for this objective. The review is methodically structured and logical, encompassing detection techniques, biometric elements of the sensors, the application of modified materials and various detection methods. Nevertheless, it is imperative to acknowledge certain shortcomings: 

1.      Selectivity is a pivotal feature of the sensor, with different modified materials and detection methods resulting in varying degrees of selectivity. The article uses the term "improved/increased selectivity" when referring to a particular material or method, but further clarification is required to understand the underlying physicochemical processes leading to enhanced selectivity. It is unclear whether selectivity is determined by incubation with a single analyte or by comparison to incubation with an analyte serving as a blank control. These two issues should be explicitly explained to evaluate the effectiveness of the different approaches.

2.      In addition to Figure 6, the creation of a table analogous to Table 1 is recommended in order to facilitate the systematisation of the parameters and characteristics of both implantable and non-implantable sensors.

Author Response

Reviewers' comments:

  1. Selectivity is a pivotal feature of the sensor, with different modified materials and detection methods resulting in varying degrees of selectivity. The article uses the term "improved/increased selectivity" when referring to a particular material or method, but further clarification is required to understand the underlying physicochemical processes leading to enhanced selectivity. It is unclear whether selectivity is determined by incubation with a single analyte or by comparison to incubation with an analyte serving as a blank control. These two issues should be explicitly explained to evaluate the effectiveness of the different approaches.

Response 1: Thank you for your valuable suggestion. We have expanded the explanation of how different modified materials and detection methods affect the selectivity of the sensor (lines #165-167, 257-263, 282-285, 279-285, 288-293). The physicochemical processes leading to enhanced selectivity, such as the specific interaction between lactate and immobilized enzymes (lines #160-164) or sensor surface materials (lines #200-208), are now clearly described. Additionally, we have included details on the sensor’s interference resistance: the improvement in selectivity is determined by measuring the sensor’s response to a single analyte in the presence of potential interferents, rather than by comparison with a blank control (lines #203-208, 503-505). This approach better reflects the sensor’s performance in complex biological matrices.

  1. In addition to Figure 6, the creation of a table analogous to Table 1 is recommended in order to facilitate the systematisation of the parameters and characteristics of both implantable and non-implantable sensors.

Response 2: Thank you for your valuable suggestion. We have fully taken your feedback into account and expanded Table 4 (Electrochemical Biosensor for Lactate) to include the sensors discussed in Section 6. We have organized all the sensors (both implantable and non-implantable) by the lower limit of their detection range, from low to high, to help readers better understand the performance of different material-based sensors in lactate detection. To emphasize the implantable characteristics of the sensors discussed in the review, we introduced a new column titled "Imp." (implantable), which serves as an important reference for readers. This will help readers select the appropriate type of sensor based on their specific application scenarios and sensor performance parameters, thereby enhancing their understanding of the different use cases for lactate detection.

Round 2

Reviewer 2 Report

Comments and Suggestions for Authors

The authors have responded convincingly to my comments and the changes and additions made are sufficient for the article to be published in Sensors.